# Readthrough Activators and Nonsense-Mediated mRNA Decay Inhibitor Molecules: Real Potential in Many Genetic Diseases Harboring Premature Termination Codons

**DOI:** 10.3390/ph17030314

**Published:** 2024-02-28

**Authors:** Nesrine Benslimane, Camille Loret, Pauline Chazelas, Frédéric Favreau, Pierre-Antoine Faye, Fabrice Lejeune, Anne-Sophie Lia

**Affiliations:** 1GEIST Institute, University of Limoges, NeurIT UR 20218, F-87000 Limoges, France; camille.loret@unilim.fr (C.L.); pauline.chazelas@unilim.fr (P.C.); frederic.favreau@unilim.fr (F.F.); pierre-antoine.faye@unilim.fr (P.-A.F.); anne-sophie.lia@unilim.fr (A.-S.L.); 2Centre Hospitalo-Universitaire (CHU) Limoges, Department of Biochemistry and Molecular Genetics, F-87000 Limoges, France; 3University of Lille, Centre National de la Recherche Scientifique, Inserm, CHU Lille, UMR9020-U1277—CANTHER—Cancer Heterogeneity Plasticity and Resistance to Therapies, F-59000 Lille, France; fabrice.lejeune@inserm.fr; 4Centre Hospitalo-Universitaire (CHU) Limoges, Department of Bioinformatics, F-87000 Limoges, France

**Keywords:** nonsense mutation, readthrough, premature termination codon (PTC), genetic disease, nonsense-mediated mRNA decay (NMD), translation

## Abstract

Nonsense mutations that generate a premature termination codon (PTC) can induce both the accelerated degradation of mutated mRNA compared with the wild type version of the mRNA or the production of a truncated protein. One of the considered therapeutic strategies to bypass PTCs is their “readthrough” based on small-molecule drugs. These molecules promote the incorporation of a near-cognate tRNA at the PTC position through the native polypeptide chain. In this review, we detailed the various existing strategies organized according to pharmacological molecule types through their different mechanisms. The positive results that followed readthrough molecule testing in multiple neuromuscular disorder models indicate the potential of this approach in peripheral neuropathies.

## 1. Introduction 

It has been estimated that many genetic diseases can be caused by a premature termination codon (PTC) within coding genes [1]. According to the National Organization for Rare Disorders (http://www.rarediseases.org, accessed on 13 February 2024), 2400 distinct genes among the 7000 genes responsible for rare genetic diseases existing in the human population are sensitive to the presence of PTCs. Amid those diseases, we could name cystic fibrosis (CF), which results from an alteration within the *CFTR* gene (cystic fibrosis transmembrane conductance regulator); Duchene muscular dystrophy (DMD), characterized by progressive muscle degeneration caused by mutations in the *DMD* gene; and many cancers often linked with *TP53* mutations. Also, multiple neurodegenerative diseases [2] are found within these 2400 PTC-triggered rare genetic disorders, often leading to a serious and progressive condition observed in various age onsets.

Charcot–Marie–Tooth disease (CMT) is a peripheral disease with a prevalence of 1 in 2500 cases. Also known as sensory–motor neuropathy, this disorder affects both motor and sensory nerves. The first-onset symptoms affect early childhood to late adulthood, typically with muscular atrophy in the feet and/or hands and weakness in the limb muscles [3]. Nonsense alterations in many CMT-associated genes result in a PTC, as has been previously shown in the *GDAP1* gene, among others [4]. Nonsense mutations have been associated with both axonal and demyelinating forms of neuropathy [5]. Indeed, sequencing performed on 17,880 patients suffering from CMT associated with a panel of 14 genes, reported in the case of *SH3TC2* a very high majority are nonsense mutations, some of which were found to be pathological [6]. 

Amyotrophic Lateral Sclerosis (ALS) is a neurodegenerative disease characterized by the progressive loss of motor neurons (MNs). The most common age of disease onset varies between 40 and 70 years old. In some cases, mutations inducing premature stop codons in the gene encoding the Cu/Zn superoxide dismutase (*SOD1*) have been reported to be responsible for this disorder. 

Frontotemporal Dementia (FTD) refers to a group of heterogeneous neurodegenerative dementias characterized by frontal and temporal brain deteriorations leading to behavioral troubles and language disorders [7]. Pathogenic mutations have been detected in FTD patients 40 to 65 years old, especially in the *PGRN* gene, whose main role is to regulate the production of progranulin [8]. Most of them are classified as nonsense mutations, resulting in lysosomal dysfunction [9].

### 1.1. Origin of PTC

Just as with natural stop codons, UAA, UAG, and UGA are the three types of codons that are able to give rise to PTCs. The emergence of such premature stop codons in mRNA can occur at different levels (DNA, RNA) related to gene expression [10].

#### 1.1.1. DNA Level

PTC can be introduced into a DNA molecule through DNA replication or transcription processes, resulting in frame-shift or nonsense mutations [1]. A frame-shift mutation generally refers to the deletion or insertion of a number of nucleotides not divisible by three, changing the reading frame and perturbing the genetic translation. Meanwhile, nonsense mutations are defined as the substitution of a single DNA base pair, and, according to a meta-analysis of the Human Gene Mutation Database (HGMD), these account for approximately 11% of alterations affecting gene-coding regions [2]. It has been reported that 23 different nucleotide substitutions affecting the gene-coding region could give rise to a PTC. Of the three nonsense codons, TAG (40.4%) and TGA (38.5%) represent the two most frequent pathological nonsense mutations reported from 995 genes, with approximately the same ratio, while the nonsense codon TAA (21.1%) is less commonly described. According to this same meta-analysis [11], the conversion of CGA (Arg) to TGA and CAG (Gln) to TAG represents the highest proportion of PTCs causing transition. The Predominantly observed substitution of C for T (44%) results from the deamination properties of 5-methylcytosine within CpG sites, which converts cytidine into uracil, which is later corrected in thymidine [11,12].

#### 1.1.2. RNA Level

mRNA-carrying PTCs can be the result of errors occurring during the transcription or splicing processes [10,13]. Splicing mutations occur within an intron or an exon and result in the creation of new splice sites or in the disruption of existing ones. Transcription is a high-fidelity mechanism through which errors occur only 0.05 to 0.5% [10] of the time compared to alternative pre-mRNA splicing; a third of mRNA isoforms obtained by alternative splicing are degraded by nonsense-mediated mRNA decay (NMD) due to the presence of a PTC [14]. 

In the past few decades, various therapeutic approaches have been developed to counteract the negative effects of PTCs. Some of them, known as nucleic acid-based approaches, exploit different molecular therapy strategies, such as tRNA suppressors, antisense oligonucleotides, and genome editing [15]. Here, we will focus on two alternative methods based on small-molecule drugs [16]: readthrough molecules and NMD inhibitors (NMDIs). Several studies have already demonstrated the effectiveness of nonsense restoration using these pharmacologic drugs. 

## 2. Natural Translation Mechanism

To synthesize a protein, a cell needs the corresponding genetic code, also known as DNA. First, the nucleotide sequence is transcribed and processed into mRNA, and then, mRNA is subjected to translation into an amino acid chain, forming the protein of interest. mRNA is therefore an essential molecule for the proper functioning of cells since it establishes the link between genetic information and protein production. mRNA is made up of a series of nucleotides which, by triplet, correspond to a particular codon. Each of the 61 existing codons are associated with one of the 20 amino acids. With each codon composing the mRNA, the ribosome is associated with a particular aminoacyl-tRNA anticodon. 

The ribosome is composed of two subunits made of protein components and ribonucleic acid (rRNA). In eukaryote cells, the 80S ribosome consists of the 60S large subunit and the 40S small subunit with 18S RNA and 33 proteins, while in prokaryote cells, the 70S ribosome consists of the 50S large subunit and the 30S small subunit with a 16S RNA subunit and 21 proteins. Three main active binding sites where mRNAs and tRNAs interact are present at the interface of these two subunits. The A site (for Aminoacyl) corresponds to the first location of the amino-acylated tRNA, carrying an amino acid bind when entering the ribosome, the P site (for Peptidyl) detains the tRNA carrying the current peptide chain synthesis, and finally, the E site (for Exit) is where the deacylated tRNA resides, ready to leave the ribosome. 

### 2.1. Ribosome Fidelity: The Role of the Decoding Center 

The ribosome holds the main role in translational fidelity. The decoding center, situated on the small ribosomal subunit at the interface with the big one, is involved in the recognition and faithful selection of the codon–anticodon tRNA. The distinction between cognate tRNA and near-cognate tRNA happens when mRNA codons are positioned at the A-site, located within the small subunit. Among the 20 aminoacyl-tRNA (aa-tRNA), only the cognate one is to be presented at the A-site. We may wonder how the ribosome manages to discriminate high-fidelity cognate tRNAs from near-cognate (single mismatch with the mRNA codon) or non-cognate (more than one mismatch with the mRNA codon) ones.

During the translation step of the elongation, the process of aa-tRNA selection is monitored by two distinct steps: the initial selection and the proofreading. The latter is driven by the energy produced following GTP hydrolysis right after the initial selective binding [17].

Noncognate tRNAs are rejected during the first step, while near cognate tRNAs that escape the first selection end up rejected during the second one. The accuracy of tRNA selection by the ribosome alone is not enough to discriminate noncognate/near-cognate codon–anticodon interactions; other elements of monitoring present in the decoding center are necessary to adjust the ribosome accuracy [3]. 

At the atomic level, a previous foot-printing experiment allowed researchers to identify significant regions encompassing the A-site, and in particular, the two universally conserved adenines at positions 1492 and 1493 (prokaryote numbering) [18], A1755 and A1756 (eukaryote numbering) [19]. Helix 18 (H18) also encompasses an important region, as it carries the conserved G530 base (Figure 1). These nucleotides are essential for the binding of the cognate tRNA at the A-Site, since they play a fundamental role in the discrimination between correct and incorrect codon/anticodon pairings, as consecutive adenines have a strong preference for canonical Watson–Crick pairs. 

When no ligand is bound to the ribosomal A-site, residues A1492 and A1493 are positioned in the internal loop of helix 44 (H44). When cognate tRNA binds to their codon at the A-site, both the A1492 and the A1493 residues are displaced outside helix loop 44, altering its conformation from an “off” state to an “on” state. The observed conformational change is necessary to allow A1492 and A1493 to interact specifically with the base pairs formed by the cognate codon–anticodon interaction, therefore stabilizing preferentially the minihelix formed between cognate tRNAs and mRNA codons. At the same time, the G530 base undergoes changes from a *syn* conformation to a *trans* one. G530 therefore interacts with A1492, just like the second position of the anticodon and the third base of the codon. The same phenomenon is observed with A1755 and A1756 in eukaryotes [19].

### 2.2. Course of Natural Translation Termination Mechanism

Physiologically, during the translation, the ribosome moves along the mRNA by incorporating the corresponding amino acid until it reaches the natural stop codon (NTC). When one of the three stop codons, UAA, UAG, or UGA, occurs through the ribosomal A-site, it will not be recognized by any existing tRNA, but by a release factor (RF). 

In eukaryotic cells, translation termination is managed by two factors: eRF1 and eRF3. The first one recognizes the three-termination codon at the A-site, then forms a complex with eRF3, which possesses a GTPase activity. The hydrolysis of GTP provides the energy that allows the cleavage between the polypeptide chain and the tRNA to which it is attached and thus releases the polypeptide chain [20] (Figure 2A), allowing the production of the normal protein. 

To enhance the translation termination efficiency, the ternary termination complex (eRF1-eRF3-GTP) requires other stimulation, notably that provided by the complex polyA-binding protein (PABP) located at the 3′UTR of mRNA [20].

## 3. Disturbance of the Translation Mechanism: Consequences of PTCs within the mRNA 

The presence of a PTC in the mRNA sequence prevents the ribosome from continuing the translation process up to the end of the open reading frame, leading to the recruitment of the release factors eRF1 and eRF3, which favors translation termination (Figure 2B). This disturbance results in either the production of a truncated protein or the degradation of the altered mRNA by the nonsense-mediated decay (NMD) system. 

### 3.1. Truncated Protein: The Aftermath 

The presence of a PTC in the mRNA sequence might lead to three main consequences. First, (i) the predicted truncated protein could accumulate in the cell, leading to high toxicity. The regular effect of such a protein could become stronger; furthermore, a new abnormal function may appear. This phenomenon is also known as “gain-of-function”. In ALS disorder, one hypothesis is that PTCs in SOD1 mRNA could lead the protein to adopt a new conformation. This change would likely enable the misfolding of SOD1, and as a consequence, its interaction with other molecules, which could lead to the formation of aggregates, toxic for the cells [21]. 

Besides the gain-of-function mechanism, the existence of a PTC within the mRNA could induce a (ii) “dominant negative effect” in which the unfunctional truncated protein could interfere in *trans* with the wild type one and subsequently void its function [22]. This mechanism was shown in β-thalassemia models, when the PTC was located on the last exon, and could escape from the NMD mechanism [23]. Finally, (iii) “loss-of-function” is the third of the main consequences that could arise. In the event of disorders transmitted as a recessive trait, as in *GDAP1*-associated recessive CMT neuropathy, the predictable truncated protein carrying a mutation would be totally deprived of its function. Due to this loss of function, calcium homeostasis and mitochondria–endoplasmic reticulum interaction would be altered [24,25]. Also, loss of function could arise as a consequence of haploinsufficiency. This mechanism is exclusively found when the disorder is transmitted as a dominant trait, as was previously shown in Frontotemporal Dementia [8]. This phenomenon occurs when one copy of a gene becomes unfunctional and the remaining functional copy cannot solely preserve the normal function.

### 3.2. Degradation of Altered mRNA by the NMD System

Cells are able to deploy natural mechanisms to protect themselves from PTCs’ adverse effects. We will focus on one of them, known as the NMD system. PTC-containing mRNA does not necessarily lead to truncated protein production, as they can potentially be recognized and degraded by this system. NMD is an mRNA-, cytoplasm-, and translation-dependent quality control mechanism, conserved across all eukaryotic cells. The pivotal role of this system concerns the recognition and the degradation of aberrant mRNAs harboring a PTC. Therefore, the NMD system prevents the production of truncated or erroneous proteins that could have deleterious effects for the organism. 

In mammalian cells, the NMD can discriminate between NTC and a PTC based on the presence of a multi-subunit protein complex (eIF4A3, MAGOH, Y14, MLN51) named exon junction complex (EJC) [26]. Amid the pre-mRNA splicing, EJCs are laid by the spliceosome, a protein complex that precisely cuts out introns from pre-mRNA, 20 to 24 nucleotides (nt) upstream of the exon junction complex. EJCs accompany mRNA from the nucleus to the cytoplasm where the single-stranded RNA is translated [27,28]. 

In physiological circumstances, the force of the ribosome during the pioneer round of the translation process is sufficient to remove all EJCs carried by the mRNA until the NTC. This process ensures that no EJCs remain on mRNA up to the translation termination. However, if the ribosome displacing the EJCs encounters a PTC, its progression is disrupted and the EJCs’ removal ceased. The remaining EJC signals the presence of the PTC located typically >50–55 nucleotides upstream of the last exon–exon junction and leads the abnormal mRNA to its degradation. The resulting discharge of the ribosome from the mRNA transcript provides enough time to recruit NMD factors [29] (Figure 3). 

#### 3.2.1. NMD Factors

Genetic screens in *Saccharomyces cerevisiae* and *Caenorhabditis elegans* were used to identify conserved factors that constitute the core machinery of the NMD system. In human cells, activation of NMD includes UPF proteins (upframeshift) and SMG proteins (suppressor with morphological effect on genitalia) [30]. See below for a brief description of each of the NMD core factors to clarify their different functions involved in the NMD mechanism conserved in human cells [31] (Figure 4). 

#### 3.2.2. NMD Mechanism Course

To achieve the degradation of mRNAs containing an early stop codon, UPF1, considered as the main key factor, triggers the NMD mechanism and undergoes phosphorylation and dephosphorylation cycle steps, which are essential for NMD progression [32]. This phenomenon intervenes exclusively when the translation termination takes place at a PTC [33,34].

Briefly, when the ribosome is stalled at the PTC position, UPF1 binds to the translation termination complex, composed of eRF1 and eRF3, already located at the PTC position. This complex, together with UPF1, associates with the SMG1 kinase to form the surveillance complex named SURF [35]. Thanks to the recruitment of UPF2 and UPF3b at the EJC position, a bridge is formed to join UPF1 to the mRNA, allowing the whole SURF complex to interact with UPF2 (via its CH domain) and UPF3b, as well as with the EJC complex to form the decay-inducing complex (DECID) [36]. Stimulated by DECID formation, phosphorylation of UPF1 is carried out by the SMG1c complex, comprised of the protein kinase SMG1, and two additional sub-units, SMG8 and SMG9. SMG9 associates tightly with SMG1, then SMG8 binds to the preformed SMG1c complex. All three factors regulate UPF1 activity through the induction of conformational changes, impacting its phosphorylation state [35,37]. Recently, another kinase named AKT1 was identified to be involved in the phosphorylation of UPF1 [38,39].

mRNA degradation can proceed through several pathways. Indeed, subsequent to its phosphorylation, UPF1 induces the recruitment of either the endonuclease SMG6 or the SMG5-SMG7 complex that participates in mRNA decay and prevents the resumption of the translation. SMG6 cleaves the mRNA in the vicinity of the PTC, whereas the SMG5-SMG7 complex not only recruits both a decapping and a deadenylation complex but also triggers the dephosphorylation of UPF1 through recruitment of the protein phosphatase 2A (PP2A). mRNA degradation occurs at both 3′ and 5′ ends [40]. After its deadenylation, the 5′ to 3′ end will be the target of exoribonucleolytic enzyme XRN1; meanwhile, the 3′ to 5′ end is exposed to degradation by the exosome [28] (Figure 5). 

### 3.3. Impact of the PTC Position in the NMD System and Illustration in Peripheral Neuropathy

The PTC location within the mRNA sequence influences the NMD system carrying out such mRNA degradation, therefore affecting the severity of the phenotype. In the neurological phenotypes, the position of the nonsense mutation seems to hold a significant impact depending on the disorder. It is the case for the *SOX10* gene. When the PTC is located in the last exon, the NMD system is unable to recognize it. This occurrence induces the production of a truncated protein, leading to a more severe form of the disorder, then called PCWH, which includes four complex syndromes: peripheral demyelinating neuropathy, central demyelinating leukodystrophy, Waardenburg syndrome, and Hirschsprung disease [41]. On the contrary, if the mRNA-PTC is subject to the NMD system, the phenotype is less severe, leading to the development of only some of the symptoms [41]. 

## 4. Readthrough Mechanism

As previously mentioned, natural translation termination is elicited by the ternary termination complex (eRF1-eRF3-GTP) at the NTC. However, several mechanisms of translation termination suppression exist. Among them a process called stop codon “readthrough”, enabling the translation of the stop codon [42], thus driving the nonsense codon to be read as a sense one, in the same reading frame [42], until the next termination signal. However, this phenomenon remains quite rare, with a frequency < 0.1% [43]. 

### 4.1. The Translational Readthrough

The translational readthrough process was first discovered in both E. Coli bacteria and in the tobacco mosaic virus (TMV). In the latter case, the tobacco virus was able to produce two polypeptide chains of 126 KDa and 186 KDa that resulted from the readthrough of an UGA codon by incorporation of a non-cognate tRNA AUG (Tyr) [44]. In this case, translational readthrough was shown to be essential for both the viability of the virus and the control of its replication level. Translational readthrough has now been identified in many viruses [45,46,47], yeast [44], Drosophila [48], and mammalian models. More recently, readthrough has been identified in a few human genes as a mechanism meant to regulate their expression by producing two isoforms from a single mRNA [49]. One of the rare cases of translational readthrough in humans concerns the transmembrane protein MPZ (myelin protein zero), which accounts for approximately 50% of the peripheral myelin protein content. The readthrough phenomenon led to C-terminally extended myelin zero (L-MPZ) proteins, which could still be involved in myelination [40]. Also, in a second case, natural readthrough has been shown to provoke the addition of peptide extensions to the sequence encoding human vascular endothelial growth factor A (VEGF-A), generating a VEGF-Ax (x: extended). This extension allowed the stop codon to be decoded into a serine one, which led to a change in its activity from pro-angiogenic to anti-angiogenic [50]. 

### 4.2. Readthrough Therapeutic Approach

As part of a promising therapy for the treatment of neuropathies resulting from an alteration inducing a PTC, natural or designed small molecules conferring a readthrough potential need to be strongly considered. Such components could allow the ribosome to pass through the PTC and resume translation until the NTC, inserting a stand-in amino acid in place of the one originally constituting the PTC (Figure 2C). The new protein may differ by only one amino acid from the wild type one, leading to a possible missense mutation. Nonetheless, such a protein could still offer a less severe phenotype [41], as is suggested in the case of the *GDAP1* gene. Indeed, CMT patients carrying a homozygous nonsense mutation in *GDAP1* develop a more severe form of the disease compared to patients harboring a missense mutation.

When the PTC passes through at the ribosomal A-site, competition between tRNA near cognate and the complex of translation termination eRF1/eRF3-GTPase will arise. Certain events, like the nature of the stop codon, the nucleotide sequence in the vicinity of the PTC post-transcriptional modifications, or small drugs, could promote an influence on the readthrough mechanism (Figure 2C).

As detailed above, the natural readthrough process inserts an amino acid into the polypeptide chain at the PTC during translation. Researchers hijacked this mechanism as a therapeutic approach to insert a near-cognate amino acid at the PTC level. This paved the way for the search for molecules activating the readthrough system to correct a disorder induced by nonsense mutations. In this review, we classified the readthrough activating molecules according to their mechanism of action into three main categories: Ribosome-targeting molecules;tRNA post-transcriptional inhibitors;eRF1-targeting molecules.

#### 4.2.1. Ribosome-Targeting Molecules

➢
**Aminoglycoside**



**Discovery**


Aminoglycosides are natural compounds synthetized by microorganisms. The first aminoglycoside ever discovered was streptomycin in 1944. This component was previously used as an antibacterial agent against *Mycobacterium Tuberculosis* [51]. Following this discovery, several drugs naturally produced were isolated from soil bacteria, including neomycin, kanamycin, paromomycin, gentamicin, amikacin, and geneticin (also known as G418) [52,53]. Most aminoglycosides are known as antibacterial components, mainly prescribed to treat Gram-negative bacterial infections.


**Chemical structure**


Aminoglycosides are characterized by a common central ring, called 2-deoxystreptamine (2-DOS), which represents ring II. This ring can be linked to a variety of amino sugars by glycosidic linkage. According to the carbon substitution position, aminoglycosides can be classified into mono-substituted aminoglycosides (substitution at C4 position, referring to ring I) and di-substituted ones (substitution at both the C4 and C5 positions or at both the C4 and C6 positions). The substituent at the C5 or C6 positions of the 2-DOS is referred to as ring III [54] (Figure 6).


**Mechanism of action**


Aminoglycosides are polar polycationic compounds that strongly bind to the ribosomal decoding center (A-site) located in the small ribosomal subunit in both prokaryotes and eukaryotes. In light of the A-site role, Aminoglycoside’s mechanism of action is mainly centered on the loss of translation accuracy and fidelity [56].


**Aminoglycosides’ interaction with the prokaryote’s ribosome**


Several studies based on crystallography and Nuclear Magnetic Resonance (NMR) have helped to understand how aminoglycosides stimulate the misreading of the genetic code to allow the acceptance of near-cognate codon–anticodon interaction [57,58]. One of the most studied structural models concerning the interaction of aminoglycosides with the bacterial ribosome relates to paromomycin [57,58,59]. In the presence of paromomycin, the 2-DOS ring strongly binds to the A1408 residue in the 16S rRNA of the A-site just across the A1492 and A1493 residues. This tight link induces residues A1492 and A1493 to bulge outside of the internal loop of helix 44 in an “on” state [57]. Then, the bacterial ribosome loses its ability to discriminate between cognate and non-cognate tRNA–mRNA associations, thus introducing dysfunctions in the protein sequence during the elongation of the translation, leading to the inhibition of protein synthesis and to bacterial death, hence its antibiotic effect [60] (Figure 7A). 


**Aminoglycosides’ interaction with the eukaryote’s ribosome**


Using X-ray crystallography and single-molecule FRET (smFRET) imaging, Prokhorova et al. highlighted that G418 (4,6-linked aminoglycoside) could interact with the 80S ribosome of *S. cerevisiae* [61]. At the molecular level, aminoglycosides bind to the canonical eukaryotic ribosomal decoding center 18S rRNA and displace the universally conserved nucleotides A1755 and A1756 outside of helix 44 [61]. The link is weaker than the one displayed in prokaryotes, which results in the progression of the translation. The difference in readthrough efficiency of aminoglycosides between eukaryotes and prokaryotes can be explained by the presence of a guanosine nucleotide in eukaryotes’ 18S rRNA at position 1645, responsible for the weak affinity of aminoglycosides to the mammalian ribosomal 18S [61] (Figure 7B). 

Due to their ability to interact with ribosomal RNAs during the translation process, aminoglycosides can be used as drugs to potentially treat several human genetic disorders caused by nonsense mutation [62]. Aminoglycosides are the first- and the best-characterized drugs that enhance PTC readthrough. Not all aminoglycosides induce PTC translation with the same efficiency. Notably, aminoglycosides containing ring I at the 6’-OH group, such as paromomycin, are more efficient at promoting the readthrough of PTC compared to aminoglycosides containing a 6’-NH2 group at this same position [16,61]. 


**Aminoglycosides as a readthrough inducer in various models**


For many cancers, as well as for various neuromuscular and neurodegenerative disorders caused by nonsense mutations, aminoglycosides represent, thanks to their ability to promote PTC readthrough, a promising therapeutic approach [62]. Thus, many studies have demonstrated the capacity of aminoglycosides as readthrough molecules in different genetic disease models in vitro and in vivo, until clinical trials.

In 1985 and for the first time, Julian and Mogg published their results concerning the phenotypic suppression of nonsense mutations after aminoglycoside (paromomycin and G418) treatment. In this study, they treated mammalian cells (Cos-7) with various doses of paromomycin and G418. Cos-7 cells had previously been transfected with a plasmid containing a bacterial gene comprising an amber nonsense (TAG) codon at position 38. With either treatment, the readthrough of this mutation was restored up to 20% levels of wild type protein [63]. Later on, in 1996, the efficacy of G418 and gentamicin was demonstrated in Hela cells. In this study, cells were transfected with a plasmid vector harboring either the nonsense mutation p.Gly542* or p.Arg553*, inducing a UGA mutation in the *CFTR* gene. Alteration in this gene is at the origin of cystic fibrosis development. Aminoglycosides were also evaluated in neurological diseases, such as Ataxia–Telangiectasia (A-T), that encode the *ATM* gene. In this model, various lymphoblastoid cell lines derived from A-T patients harboring different PTCs were treated with geneticin and gentamicin. Following the treatment, cells were able to restore the synthesis of functional ATM protein [64].

The *Mdx* mouse model for Duchenne Muscular Dystrophy (DMD) is the first animal model for which the efficacy of aminoglycosides has been tested. *Mdx* rodents underwent subcutaneous injections of gentamicin once per day for 14 days. After the treatment, expression of the dystrophin protein was restored up to 10 to 15% of the level found in wild type mice. This therapeutic approach also promoted a significant improvement of the muscle function [65]. However, the beneficial results of gentamicin treatment in the *mdx* model could not be replicated by other teams [66]. Nonetheless, a second positive result was observed in a different mouse model created by Ming Du et al. [67]. They reported that *transgenic* mice carrying a human nonsense mutation (p.Gly542*) in *CFTR* (*Cftr−/−*) exhibited re-expression of functional CFTR under gentamicin (14%) or tobramycin (5%) through daily administration [67]. 

Two previous encouraging pilot studies set up by Wilschanski et al. [68] and Clancy et al. [69] investigating the consequences of gentamicin treatment in *CFTR* patients were quickly supported by a randomized, double-blind, placebo clinical trial. This third study reported that gentamicin therapy administrated during two successive weeks in 19 CF patients carrying a PTC (in either an homozygote or heterozygote state) led to the production of full-length CFTR protein [70]. A total of 25% and 35% of recovery were observed in patients carrying the p.Arg553* and p.Gly542* mutations, respectively. Notably, in another clinical trial, daily intravenous gentamicin administered to four patients suffering from DMD and Becker Muscular Dystrophy carrying various stop codon sequences did not permit any dystrophin re-expression [71,72,73].

Later on, in 2016, five patients suffering from Recessive Dystrophic Epidermolysis Bullosa, an incurable disease caused by mutations in the gene encoding type VII collagen, entered into another aminoglycoside PTC correction therapy clinical trial. In this study, patients harboring various nonsense mutations (p.Arg236*, p.Arg2814*, p.Arg578*, p.Arg613*, and p.Arg683*) in the *COL7A1* gene were treated with topical and intradermal administration of gentamicin for 3 months. A recovery varying from 20% to 165% of the level of functional type VII collagen protein was obtain and persisted for 3 months [74]. Topical gentamicin administration also corrected dermal–epidermal separation, improved wound closure, and reduced blister formation in the treated patients. 

In addition, a single patient suffering from both Epidermolysis Bullosa Simplex and Muscular Dystrophy (EBS-MD) carrying a nonsense mutation in *PLEC1* underwent aminoglycoside treatment. EBS-MD is an autosomal recessive disorder caused by pathogenic variants in *PLEC1*, which encodes plectin protein. The patient received gentamicin (7.5 mg/kg/d) for 14 consecutive days in July 2019 and February 2020. Following the treatment, expression of plectin in the skin was detected for at least 5 months [75], and the patient also showed signs of both skeletal and respiratory muscle improvement. 


**Aminoglycoside side effects**


Despite the readthrough efficiency displayed by many aminoglycosides in several genetic disease models, those molecules are also well-known, after long-term treatment, to induce side effects such as ototoxicity, nephrotoxicity, and to a lesser extent, retinal toxicity, which limits their clinical use as a PTC correction therapy [76,77]. These adverse effects can partially be explained by the selective accumulation of aminoglycosides in the kidney and in the cochlear hair cells of the inner ear [76,77]. Indeed, previous studies suggested that the nephrotoxicity following accumulation of aminoglycosides in the kidney could result in their binding to the large glycoprotein receptor named LRP2 (also known as megalin). This receptor, mainly present at the apical surface of absorptive epithelia, such as in the kidney, mediates the uptake of aminoglycosides into the cell by endocytosis. Also, aminoglycosides were described to be endocytosed at the apical membranes of hair cells and transported to lysosomes. Such lysosomal sequestration, with accumulation, was hypothesized to induce lysosomal lysis, releasing aminoglycosides in the cytoplasm. Such adverse effects can also be explained by the role of aminoglycosides in ROS generation. Indeed, the positive charge carried by aminoglycosides facilitates their interaction with several negatively charged components of the cell, such as phospholipids and phospholipases. This property allows aminoglycosides to bind to the A site of the mitochondrial eukaryote ribosomal 12S rRNA, which shares similarity with the A site of the bacterial ribosome [78]. Thus, mitochondrial protein production could be hindered, leading to mitochondrial dysfunction and ROS generation, resulting in cell death [78,79,80,81].


**Molecules capable of enhancing aminoglycoside activity**


Interestingly, some studies reported that G418 PTC readthrough activity could be enhanced by other compounds, reducing G418 treatment concentration. Those molecules are referred to as CDX molecules and Y-320. Among the CDX chemical molecules identified, CDX5 displayed the best readthough activity [82]. CDX5 and G418 have shown high readthrough capacities when tested on a neuronal model derived from human induced pluripotent stem cell (hiPSc)-bearing nonsense mutations in the progranulin gene (*PGRN*) [82]. 

Recently, a new small readthrough molecule (SRTM) known as Y-320 was identified. Alone, this molecule did not permit any readthrough efficacy. In contrast, co-treatment with G418 and Y-320 increased PTC readthrough at a higher level than G418 alone [83]. Study on the Y-320 mechanism highlighted that this component could increase ribosome biogenesis, improving protein production. 

To overcome side effects that hampered the potential of aminoglycoside read-through therapeutic approaches, derivatives were chemically designed, separating structural components with readthrough properties from those affecting cell viability [84,85,86].


**Development, chemical structure, and mechanism**


Structural modifications of certain aminoglycosides generated three classes of derivatives: TC derivatives, designed from neomycin; JL derivatives, designed from kanamycin; and B and NB derivatives, designed from paromomycin or G418 [87]. The modification of the paromomycin structure led to the development of the first NB generation, named NB30. They preserved the pseudo-trisaccharide structure, as it is the main recognition element for the rRNA. The pseudo-trisaccharide is used as the main structure on which other designed chemical structures can be attached [88]. Four other generations were developed later on. The second generation, named compound NB54, was designed after the introduction of an AHB group (4-amino-2-hydroxy-butanoyl) at the position N-1 of paromomycin ring II. The third one was obtained by modifications of G418. This aminoglycoside contains a (R)-6′ methyl group on the glucosamine ring (I ring), giving it the highest reading efficiency among all aminoglycosides. This (R)-6′ methyl group was used to generate the third generation by adding it on ring I of NB30 and NB54 to form NB74 and NB84, respectively. The fourth generation, called NB124, was developed after including a 5′-(S)-Me group on ring III, allowing NB124 to bind cytoplasmic ribosomes [86]. The last generation, made to target the eukaryotic cytoplasmic rRNA A site, was recently synthesized. An additional hydroxyl group was added on the side chain of the glucosamine ring I of G418, producing new molecules NB156 and NB157 [89].


**Aminoglycoside derivatives as a readthrough inducer studied in various models**


In vitro, NB30 showed mutation-correcting activity on the nonsense mutation p.Arg31* on an Usher syndrome cellular model [90]. NB54 enhanced nonsense mutation readthrough activity on various genetic diseases such as Usher syndrome, CF, DMD, and Hurler syndrome, observed in cellular models at a level at least twice as high as gentamicin, paromomycin, or NB30 [91]. In the same cellular models and compared to gentamicin, the new synthetic drugs NB74 and NB84 displayed superior readthrough efficiency and reduced toxicity [92]. 

NB124 derivative restored full-length CFTR for three CFTR UGA mutations (p.Gly542*, p.Arg1162*, and p.Trp1282*). Also, the chloride transport function was restored at 7% of its wild type level in primary human bronchial epithelial CF cells [92]. NB124 was recently evaluated in the human tumor cells HDQ-P1 (human primary breast carcinoma) carrying a nonsense mutation of p53 (Arg213*; UGA). This molecule promoted the significant production of an endogenous and functional p53 protein [85].

To determine in vivo efficacy of NB54 in Usher syndrome, day 0 postnatal mice were transfected with a plasmid carrying mutated (p.Arg31*) *USH1C* (gene encoding the scaffold protein harmonin). The transfection was performed into retinas by electroporation. After 6 weeks, mice were injected sub-retinally with NB54 (125 µg/µL). This molecule induced a recovery of the full-length harmonin, associated with high biocompatibility [93]. Also, NB124 has been tested on mice expressing a human *CFTR-Gly542** transgene in a *Cftr* knockout model. This treatment improved the mice therapeutic index by a factor of 10 compared to gentamicin, with cytotoxicity reduction [92]. 

In a clinical trial, NB124 was referred to as ELX-02 [6′-(R)-Methyl-5-O-(5-amino-5,6-dideoxy-α-L-talofuranosyl)-paromamine sulfate]. Indeed, this aminoglycoside analogue targets the ribosome with low affinity. This characteristic allowed for the increase in the specificity toward the cytoplasmic ribosome but also the decrease in the affinity for the mitochondrial one [86,94]. ELX-02 binds preferentially to the A-site of the eukaryotic ribosome, which allows significant readthrough of the UGA stop codon of *TP53* mRNA, leading to the synthesis of full-length functional protein in DMS-114 cells. In addition, it allows for the decrease in the NMD activity [95]. In patient CF-derived intestinal organoids, ELX-02 enhances CFTR expression with different mutations of the CFTR gene (Gly542*, Arg1162*, Trp679*) [96,97]. This compound was tested in a phase II clinical trial as a therapeutic approach to treat cystic fibrosis (NCT04135495) and cystinosis (NCT04069260) patients carrying nonsense mutations and was evaluated as showing good tolerance and safety for patients. Unfortunately, ELX-02 did not achieve statistical significance for CF subjects with the Gly542* mutation. Recently, a proof-of-concept trial for ELX-02 was expected to treat a rare kidney disease, “Alport syndrome”, due to its preferential absorption by the kidneys (NCT05448755). 

➢
**Ataluren (PTC124)**



**Discovery**


Following the screening of a library of 800,000 small chemical compounds using a firefly luciferase reporter gene, one molecule ((3-[5-(2-fluorophenyl)-1,2,4-oxadiazol-3-yl]-benzoic acid), also known as ataluren or Translarna, structurally different from aminoglycosides, was identified. Interestingly, ataluren resulted in low toxicity and higher readthrough activity using lower doses [98]. 


**Mechanism of action**


Considering the particular chemical structure of PTC124 compared to original aminoglycosides, a different mechanism of action was expected. Indeed, two studies highlighted that PTC124 readthrough potential, contrary to natural aminoglycosides such as G418, is derived from its ability to inhibit the release factor activity by competition. PTC124 binds with rRNA at specific sites (18S-A1195), proximal to the decoding center, but also to the peptidyl transfer center (26S-A2669, 26S-A2672, and 26S-A3093) of the ribosome [99,100]. 


**Ataluren as a readthrough inducer studied in various models**


Since its discovery in 2007 by Welch et al., PTC124’s ability to correct nonsense mutations has been the subject of several in vitro studies. At low concentrations (0.01–0.1 mM), this molecule showed the ability to promote readthrough of PTCs in primary human muscle cells from DMD patients, with higher results in UGA stop codon, followed by UAG and then UAA [98]. PTC124 has also shown its effectiveness on nonsense mutations in the *USH1C* [101] and dystrophin genes [98]. 

Recently, an in vivo study on dystrophin-deficient zebrafish has pointed out that PTC124 only displayed readthrough efficacy for the UAA stop codon. This result was pretty unexpected, as UAA is normally considered as displaying the lowest level of read-through activity [102]. The efficiency of PTC124 was also demonstrated in other mouse models for different genetic diseases: DMD [98], CF [103], Usher syndrome, and the neuronal ceroid lipofuscinoses [104].

Supported by phase I and IIb clinical trials with beneficial results in DMD patients [105], this therapeutic drug was indicated in a phase III clinical trial for the ambulatory treatment of patients aged 5 years or older. Its efficiency was evaluated on various criteria based on a walking perimeter (6 min test; 150 m) [106]. 

Other preclinical studies did not confirm the readthrough activity of ataluren. Indeed, when the drug was administrated for over 48 weeks in a cohort of 238 patients exhibiting nonsense mutation-inducing CF (phase III), there was no significant improvement in patients’ lung function [107]. This disappointing result observed in most patients could be explained by the prescription of a parallel treatment with Tobramycin. Indeed, the authors considered Tobramycin as a potential antagonist which could inhibit the effect of PTC124. However, in another phase III clinical trial in which 279 patients randomly received either ataluren treatment or a placebo, with no further prescription of Tobramycin, no significant lung function improvements were obtained [108]. 

Still, PTC124 received conditional approval by the European Medicines Agency in 2014 for the treatment of nonsense mutation Duchenne Muscular Dystrophy (nmDMD) in ambulatory patients aged 2 years and older under the trade name Translarna™.

PTC124 was reported as a nontoxic drug, orally bioavailable [98]. Currently, this is the only readthrough molecule that received global approval for a phase III test to treat cystic fibrosis and DMD diseases [106,109]. Nevertheless, after the contradictory results of a phase III clinical study, PTC124 was suspended in 2017 for the treatment of CF and DMD patients [65,106]. Besides its ineffectiveness, PTC124 was the subject of much criticism and questions, as a study indicated that the initial discovery of PTC124 may have been biased by its direct effect on the FLuc (firefly luciferase) reporter used [110]. 

➢
**PTC414**



**Discovery**


To counter the feeble and fluctuating readthrough efficiency achieved by PTC124 treatment, chemical optimization of PTC124 was performed, leading to the creation of the PTC414 molecule. PTC414 allows for maintaining the beneficial activity of PTC124 with a higher level of plasma exposure and tissue penetration, improving the pharmacokinetic characteristics in all three different PTCs [111]. 


**PTC414 as a readthrough inducer studied in various models**


Fibroblasts harboring a UAG PTC collected from humans suffering from Choroideremia (CHM), an X-linked chorioretin dystrophy due to mutations in the *CHM* gene coding for REP-1 protein, were treated with PTC414. No increase in REP1 protein was detected after the treatment [111].

PTC414 has been tested in a zebrafish model with a *CHM* gene harboring a UAA PTC *chm^ru848^* [111]. This molecule partially restored the expression of the REP-1 protein (17.2%).


**Side effect**


In a zebrafish model, after PTC414 (2 µM) treatment, signs of renal toxicity were observed. This side effect was not described after PTC124 treatment [111].

➢
**RTC13 and RTC14**



**Discovery**


To overcome the bias encountered when using the firefly luciferase gene reporter assay, a luciferase-independent HTS assay was developed for future drug screening. This alternative system exploited a protein transcription/translation (PTT) assay coupled to an enzyme-linked immunosorbent assay (ELISA) to test multiple compounds that may enhance PTC readthrough. To perform the PTT-ELISA assay, the plasmid used harbored a PTC mutation (either UGA or UAA) in the *ATM* gene, responsible for the development of ataxia–telangiectasia (A-T). The sequence carrying the PTC was N- and C-terminally tagged by two epitopes, Myc and V5, respectively. Any partially or fully translated protein would be caught on the ELISA plate thanks to an anti-myc antibody. If the compound tested promoted PTC readthrough, the concerned translated protein would be detected with anti-V5–horseradish peroxidase (HRP) antibody [112]. On the contrary, truncated protein would miss the C-ter sequence and so would not contain the V5 epitope. Consequently, the protein would not be detected by the anti-V5–horseradish peroxidase (HRP) antibody. This new method was applied to a 34,000 SRTM library. Two leading compounds were identified: RTC13 (2-imino-5-{[5-(2-nitrophenyl)-2-furyl]methylene}-1,3-thiazolidin-4-one) and RTC14 (4-tert-butyl-2-[(3-nitrobenzylidene)amino]phenol).


**Mechanism of action**


The molecular mechanism responsible for RTC13 and RTC14 readthrough activity effectiveness remains unknown. These molecules possibly display a similar mechanism of action as aminoglycoside, interfering with ribosomal translation. 


**RTC13 and RTC14 studied in various models**


Both RTC13 and RTC14 induced the restoration of the full-length dystrophin protein expression in myotubes derived from skeletal mouse muscle *mdx* carrying a UAA nonsense mutation [112].

Intramuscular injection of RTC14 compound into skeletal muscles of *mdx* mice showed no significant readthrough activity. However, intramuscular injection of the RTC13 compound led to re-expression of dystrophin in the mouse diaphragm and heart. RTC13 treatment also improved muscle function [113]. 

➢
**GJ071 and GJ072**



**Discovery**


GJ071 and GJ072 have been identified in a screening of 36,000 other molecules with the same biological system previously described to identify RTC13 and RTC14 [114]. 


**Mechanism of action**


GJ072 is known to share the same mechanism of action as PTC124 [115]. The molecular mechanism of GJ071 remains unknown.


**GJ071 and GJ072 as readthrough inducer studied in various models**


Treatment with GJ071 and GJ072 in cells derived from an A-T patient offered equal readthrough activity regardless of the three stop codons, compared to RTC13 and PTC124, that display various activity depending on the stop codon [114]. 

➢
**TLN468**



**Discovery**


In order to identify new molecules with more efficient readthrough activity, Bidou et al. used a three-step screening protocol. First, they generated a stable NIH 3T3 cell line by integrating a secreted Metridia luciferase reporter gene interrupted by a nonsense mutation Arg213* (TGA) from the *TP53* gene [116]. A live-cell assay served to screen 17,680 molecules from chemical libraries. Based on their activity to induce luciferase with a high level compared to untreated cells, 43 molecules were selected. Then, to limit the selection of false positives, a second screen was performed by a dual reporter system including β-galactosidase and firefly luciferase surrounding a PTC Arg213*. Thus, four molecules were retained. In the third assay, they demonstrated the efficacy of TLN468 to increase the mRNA level of the TP53 in human HDQ-P1 cell line, harboring the Arg213* mutation in its endogenous gene [116].


**Mechanism of action**


TLN468, a 2-guanidino-quinazoline, has been described initially as an antibacterial molecule [117], with a site of action at the level of the ribosome. However, the mechanism of action needs to be investigated. In vitro analysis reveals that TLN468 introduces cysteine for the UGA codon, glutamine for the UAG codon, and tyrosine for the UAA codon [118].


**TLN468 as readthrough inducer studied in various models**


The efficacy of TLN468 to promote readthrough of 40 different PTCs most frequently involved in DMD disease was described [116].

#### 4.2.2. tRNA Post-Transcriptional Inhibitors

➢
**2,6-diaminopurine**



**Discovery**


In order to identify new drugs with higher efficiency to treat inherited genetic diseases resulting from nonsense mutations, several natural extracts of fungi, plants, or marine invertebrates were screened. A new construct encoding firefly luciferase was used to evaluate those compounds. Its novelty lies in the fact that it carries both an intronic sequence and a nonsense mutation (TGA, TAG, or TAA) [119], therefore efficiently mimicking mRNA submission to the NMD system. This screening identified an extract from the fungus *Lepista inversa* named H7. This extract reveals luciferase activity on both the UGA and UAA stop codons with a correction almost twice as good as the one offered by the aminoglycoside G418 at 1 mg/mL [119]. Splitting of this extract revealed a new active component named 2,6-diaminopurine (DAP) [120]. 


**Mechanism of action**


It is well-known that various post-transcriptional modifications induced at the tRNA level are crucial for their structure, function, and stability, especially at position 34 of the tRNA anticodon. Among those modifications, the 2′-O-methylation of the cytosine 34 induced by the 2′-O-methyltransferase (FTSJ1) in humans, or TRM7 in yeast, is strongly involved in the fidelity of the codon recognition. It is possible to exploit these essential modifications using 2,6-diaminopurine-like inhibitors, such as DAP, to decode mRNAs disrupted by a PTC (UGA) [120]. Indeed, this molecule hinders the cytosine 34 methylation (Cm_34_) of tRNA by FTSJ1 enzymes, hampering the post-transcriptional modification process (Figure 8).


**2,6-diaminopurine-like inhibitors as a readthrough inducer studied in various models**


DAP was tested on three cancer cell lines, Calu-6 cells, Caco-2 cells, and Caov-3 cells, harboring three different endogenous nonsense mutations in the *TP53* gene, UGA, UAG, and UAA, respectively. Interestingly, DAP (25 µM) was more efficient than G418 at 1 mg/mL. In particular, for UGA nonsense mutations, it allowed the restoration of the function of p53, increasing transcriptional activity on its target gene.

The efficiency of DAP to correct nonsense UGA mutations was confirmed in immunodeficient nude mice injected with Calu-6 cells carrying a UGA nonsense mutation in the *TP53* gene. After 5 weeks of treatment with either DMSO (vehicle) or DAP (1mg per day) every day, tumor growth was significantly decreased with DAP treatment compared to DMSO. Recently, the efficacy of DAP was demonstrated in several models of CF pathology: in animals, patient-derived organoids, and patient cells carrying a UGA as a PTC in the *CFTR* gene. The pharmacokinetic analysis demonstrates a stable molecule in plasma, with high biodistribution in different tissues like lung, brain, and muscle. In this study, DAP was more effective than G418 or ELX-02 [121].

➢
**NV derivatives**


In another recent study, three small molecules named, NV848, NV914, and NV930 were shown to promote the readthrough activity by inhibition of FTSJ1, a tRNATrp-specific 2′-O-methyltransferase [122].

NV848, NV914, and NV930 are synthetized molecules containing 1,2,4-oxadiazole [123]. In vitro testing showed a higher readthrough activity in CF model systems [123,124], NV848, NV914, and NV930 were evaluated in CF mouse models with a good tolerability and without any mortality at 2000 mg/kg of NV914 and 300 mg/kg of NV848 or NV930 [125]. Further studies are necessary to support these beneficial effects. 

#### 4.2.3. Molecules Targeting eRF1

➢
**SRI-37240 and SRI-41315**



**Discovery**


In order to identify molecules with higher readthrough efficiency, Jyoti et al. searched among 771,345 compounds. They developed a novel bioluminescence system based on NanoLuc, a small luciferase enzyme derived from the deep-sea shrimp Oplophorus gracilirostris. Compared to other luciferase such as Firefly or Renilla, limited by their size, stability, and luminescence efficiency, NanoLuc reacts with the furimazine as a substrate to produce furimamide, a luminescent component, in the presence of molecular oxygen. This system offers more stability and a smaller-size luciferase (19 kDa) with high duration and luminescence intensity (>150-fold compared to firefly luciferase) [126]. This process identified two molecules named SRI37240 and SRI41315. 


**Mechanism of action**


As previously detailed, competition constantly occurs between the translation termination and the readthrough process. The readthrough efficiency depends on the capacity of the decoding center to incorporate a tRNA near cognate at the PTC position before eRF1 terminates the translation. Therefore, molecules that could deplete eRF1, such as SRI-41315, allow an extended pause at any stop codons. Hence, the crucial point is to investigate whether the NTC can also be disturbed using these molecules targeting eRF1 [127].

However, the translation termination at PTC or at NTC is different since translation termination at NTC is well-known to be more efficient. Only a high reduction in eRF1 activity at the level of NTC would be necessary to disturb the normal translation termination [127,128] (Figure 9). Thus, it has been demonstrated that depletion of eRF1 using oligonucleotides (ASO) reduced by 40% the abundance of eFR1 and promoted translational readthrough in hFIX-p.Arg338* hemophilia mice [128].


**Molecule targeting eRF1 as a readthrough inducer studied in various models**


SRI-41315 and its derivative SRI-37240 were able to rescue CFTR expression in FRT cells expressing a human gene harboring a UGA nonsense mutation in the CFTR gene compared to G418 alone. Co-treatment with SRI-37240 and G418 mediated better readthrough efficacy, resulting in 25% of wild type CFTR protein expression and function compared to the control group [127].


**Side effects**


It has been reported that SRI-37240 and SRI-41315 induce modifications on cell sodium transport [128].

### 4.3. Readthrough Limits 

Certain events, like the nature of the stop codon, the nucleotide sequence in the stop codon vicinity, as well as post-transcriptional modifications, can impact the readthrough process, either promoting or tempering it. 


**Selected amino acid incorporation in premature termination codon promoted by readthrough molecules**


In cells, the genetic code is read by complementarity between tRNA cognate anticodon and mRNA codon. However, it can be read with near-cognate tRNA, establishing two codon–anticodon bonds with a mismatch on either the third or first position of the codon and anticodon (called the wobble position). Thus, this explains how, thanks to readthrough molecules, near-cognate tRNAs can be inserted to resume translation at stop codons that normally do not match any specific tRNA. This could potentially help treat patients affected by many genetic diseases harboring PTC [129]. 

At the mispaired wobble position 3, SRTM almost exclusively allows the readthrough of UGA stop codon by preferentially inserting Trp, followed by Arg and then Cys. As for the UAG stop codon, correction was mostly performed by inserting Gln, then Tyr and Lys. Finally, with the UAA stop codon, PTCs were predominantly readthrough after insertion of Gln, followed by Tyr and rarely by Lys. As for the wobble position 1 mispairing, U-G is the most authorized readthrough of UAG which can be more easily read by the tRNA-Arg-UCG than U-U and U-C mispairing [130].

Readthrough studies using mass spectrometry revealed that treatment by 2,6-diaminopurine (25 µM) of HEK293FT cells transfected with a plasmid carrying Fluc-int-UGA led to exclusive incorporation of *Trp* at the UGA PTC position, underlying the model influence [120]. 


**Function and structure of proteins restored by readthrough process**


Promoted by the readthrough process, the incorporation of an amino acid at the PTC position could induce the production of either a native protein or could lead to the production of a modified one (varying by only one amino acid). In the latter case, investigations are always necessary to characterize the activity and the stability of the modified expressed protein. 

It was previously shown that treatment with Gentamicin in order to correct nonsense mutations in the *CD18* gene responsible for Leukocyte adhesion deficiency 1 (LAD1), an inherited disorder of neutrophil functions, was able to increase the expression of entire proteins, CD18. However, their function and subcellular localization were impaired when modified in vitro, and abnormal adhesion and chemotactic functions were also observed in vivo [131]. Those side effects were due to the replacement of Arg, the wild type codon, by Trp at the PTC.


**Factors influencing readthrough efficacy**



**The stop codon nature**


In vitro, in vivo, and clinical trial studies have been performed to evaluate the efficiency of readthrough molecule treatments on different PTCs for many genetic disorders. However, various results are linked to several factors that impact the effectiveness of readthrough, such as the PTC nature and nucleotide sequence surrounding it [132]. It is crucial to underline that readthrough molecules do not have the same influence on all three PTCs. Thus, it is well-known that pharmacological drugs are able to correct the UGA PTC more efficiently than the UAG and UAA ones, with the proposed gradual intensity: UGA > UAG > UAA [133]. The stop codon UGA differs only by the wobble position with Trp (UGG) and Cys (UGC), which increases the chances of efficient readthrough with this codon. 


**The nucleotide context**


Experimental results have shown that the identity of the nucleotide sequence upstream and downstream of the PTC could influence the translation termination efficiency.

➢
**3′context**


The presence of a pyrimidine located at the +4 position, right after the PTC (positions 1 to 3), highly stimulates a drug’s readthrough capacity. Especially, the presence of a cytosine next to the stop codon UGA and UAA and a uracil next to the UAG stop codon enhance the readthrough efficacy [42]. 

At this position, a tetranucleotides hierarchy of C > A > G > U was observed, from the nucleotide allowing the highest readthrough levels to the lowest [134]. Several studies revealed that the consensus sequence CAA, either downstream or upstream of the stop codon UAG, could influence the efficiency of the readthrough process [135,136]. 

➢
**5′context**


In addition, it was also established that the vicinity of the 5′ stop codon nucleotide context could impact the readthrough mechanism. Tork and collaborators hypothesized that the presence of two adjacent adenines (referred as positions −2 and −1) upstream of the PTC (referred as positions 1 to 3) could influence translation termination. To explain this phenomenon, it was hypothesized that the two A localized at the −2 and −1 positions of the ribosomal P site could directly bind the structural helix 44 of 18S rRNA, which then would influence the incorporation of a tRNA at the A site [137]. 

Additionally, it has been reported that the presence of the two nucleotides, upstream (uridine) and downstream (cytosine) at positions −1 and +4, respectively, of the PTC carries a decisive role in the readthrough efficiency after treatment with gentamicin [43].


**Factors influencing readthrough efficacy**


The important point that we would like to underline in the review is the question of how these readthrough molecules manage to distinguish (or differentiate) between a PTC and an NTC. Translations of PTC and NTC are two distinct mechanisms. Several hypotheses were made to explain how readthrough molecules were not able to hinder the physiological stop codon. The natural stop codon found near the poly (A) binding protein (PABP) is associated with the 3’poly (A) tail of the mRNA, which facilitate its interaction with the release factor eRF3, stimulating the efficiency of translation termination. 

However, the distance between PTC and the PABP protein decreases the interaction between PABP and eRF3, which could make the translation termination less efficient [138]. 

## 5. NMD Inhibitors 

### 5.1. Different Types of NMD Inhibitors (NMDIs)

In some cases, truncated protein synthesis could be sufficient to ensure partial or complete function of the wild type protein. Thus, the inhibition of the NMD system could be beneficial to stabilize the amount of mRNA carrying a PTC and to increase the protein synthesized [139]. The NMD system is not usually 100% efficient. Indeed, it has been reported that 5–25% of mRNA containing a PTC escaped to the NMD [140]. A reduced NMD efficiency might influence the disease severity and thus potentially save the clinical phenotype, as observed in Becker muscular dystrophy. In this case, a truncated form of the dystrophin is still synthesized in insufficient quantities, which gave a less severe phenotype than the one observed in DMD, where the dystrophin protein is totally absent [141]. 

As described previously, NMD undergoes a phosphorylation/dephosphorylation cycle of UPF1. Based on this mechanism, NMD inhibitor molecules can be classified in different categories. Among others, these are the translation inhibitor, cytoskeleton disruptors, and apoptosis inducers [142]. Here, we are going to focus on another category of NMDI: those acting on phosphorylation (caffeine and wortmannin) or on dephosphorylation of hUPF1 (NMDI 1, NMDI 14) (Figure 10). 

### 5.2. Molecules Targeting the Phosphorylation Cycle of hUPF1

➢Caffeine and wortmannin

Among NMD inhibitors, caffeine and wortmannin were investigated. Those molecules target the phosphorylation of hUPF1 through the inhibition of SMG1 kinase. Both compounds previously showed beneficial effects on fibroblasts from patients suffering from Ulrich’s disease (congenital muscular dystrophy) carrying a PTC in the collagen IV gene, resulting in extracellular matrix defects in patients [143]. Inhibition of the NMD system by treatment with caffeine or wortmannin allowed the stabilization of mRNA levels and the synthesis of a truncated form of the protein of interest. Restoration of the extracellular matrix was observed; however, their therapeutic uses were restricted due to their inhibiting activity of other kinases such as PIKKs, essential in DNA damage repair [144]. 

### 5.3. Molecules Targeting Dephosphorylation Cycle of hUPF1

➢NMDI 1 

NMDI 1 (Nonsense-Mediated mRNA Decay Inhibitor 1) was identified as the first small specific inhibitor of the NMD pathway. This molecule reduced the interaction between hSMG5 and hUPF1 [145], leading to hUPF1 hyperphosphorylation in P-bodies (eukaryote cytoplasmic structures containing some NMD factors). 

➢NMDI 14

A decade later, another NMD inhibitor, NMDI 14, was discovered. This NMDI exhibited promising results in cancer cells containing a PTC in the *P53* gene, disrupting the interactions between UPF1 and SMG7 with low toxicity [146].

➢Amlexanox 

In order to unearth new NMDIs, scientists developed cell lines able to evaluate the NMD mechanism. mRNA encoding the firefly luciferase is directed by NMD factors (one of the UPF protein) downstream of the NTC in its 3′UTR. The presence of such NMD factors downstream of the physiological stop codon of the mRNA causes the recognition of this NTC as a PTC, inducing the degradation of mRNA by the NMD. This system enabled the identification of 1200 marketed drugs, including one compound named Amlexanox (2-amino-7-isopropyl-5-oxo-5H-chromeno [2,3-b]pyridine-3-carboxylic acid). In this system, no cytotoxicity was reported from amlexanox [147].

Treatment with 25 μM of Amlexanox in cell lines from patients with cancer, DMD, and CF stabilized the amount of PTC containing mRNA. Truncated proteins were found after treatment with this NMDI; however, full-length protein of dystrophin, P53, and CFTR were surprisingly produced in several cellular models derived from patients with DMD, cancer, and CF, respectively. Therefore, Amlexanox seems to possess the ability to both stabilize mRNA containing nonsense mutation and promote PTC readthrough. Indeed, amlexanox showed readthrough activity in human cells harboring PTC mutation in *COL7A1,* from patients with recessive dystrophic epidermolysis bullosa (RDEB) [148]. In addition, amlexanox was able to stabilize *GDAP1* mRNA harboring UGA-PTC and to restore the protein expression of GDAP1 in a CMT model of hiPSC-derived neuronal cells [149]. However, the mechanism combining both effects in cells remains unclear [150,151].

### 5.4. Limits of NMD Inhibition

Additionally, regarding the quality control function of the NMD, this system also takes place in the expression regulation of more than 10% of the transcriptome. Indeed, NMD plays a crucial role in the regulation of essential biological processes such as embryonic development, cell homeostasis, cellular response to stress, regulation of the immune response, and viral replication [152]. Therefore, treatment with NMDI could potentially affect the expression of genes that are submitted to NMD transcriptomic regulation. In human cells subjected to depletion of hUPF1 or hUPF2 following NMDI treatment, transcriptomic microarray analysis of physiological transcripts revealed an overexpression of 1.5 to 4.9% of the genes analyzed [153,154].

A comprehensible fear regarding the use of NMDI concerns its impact on natural NMD substrates. However, studies displayed some fairly encouraging results. They established that Amlexanox treatment did not affect the mRNA levels of the three genes NAT9, TBL2, SC35, which are known to be sensitive to the NMD system and reductions in the amount of its translated mRNA [147,155].

## 6. By Itself, Readthrough Mechanism Is Insufficient

The cocktail combining readthrough molecules and NMD inhibitors could be an interesting strategy to treat patients with genetic diseases and cancer caused by nonsense mutations. Some molecules even display this therapeutic potential identified with the dual effect: NMD inhibition and PTC readthrough activation [82,147]. Among them, G418 harbors a high readthrough efficiency and is also an NMDI activity. Amlexanox has also been reported as being able to promote PTC readthrough and to stabilize nonsense mRNAs on three different cell lines derived from patients with CF, DMD, and lung cancer [147].

## 7. Conclusions

Aminoglycosides were the first molecules identified as readthrough inducers. Since then, several studies have determined possible new NMDI and/or readthrough activators that can correct nonsense mutations and thus partially or completely restore the protein of interest. These molecules would have the potential to radically improve the treatment of many diseases linked to nonsense mutations, especially for disorders lacking therapeutic approaches (neurological diseases, cancer, rare genetic diseases). Better knowledge and understanding of molecular mechanisms that lie behind these strategies and their variety of efficiencies according to the PTC and genetic background are essential to screen molecules that will be designed for personalized medicine.

## Figures and Tables

**Figure 1 pharmaceuticals-17-00314-f001:**
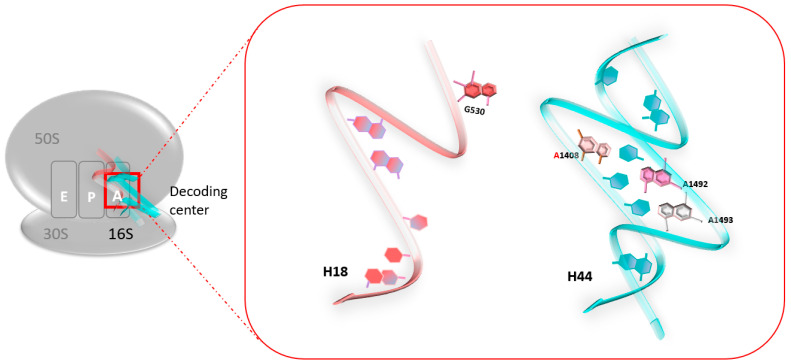
The decoding center of the 70S ribosome in prokaryotes. The two universally conserved adenines at positions 1492 and 1493 of helix 44 (H44) (blue), and conserved G530 of helix 18 (H18) (red).

**Figure 2 pharmaceuticals-17-00314-f002:**
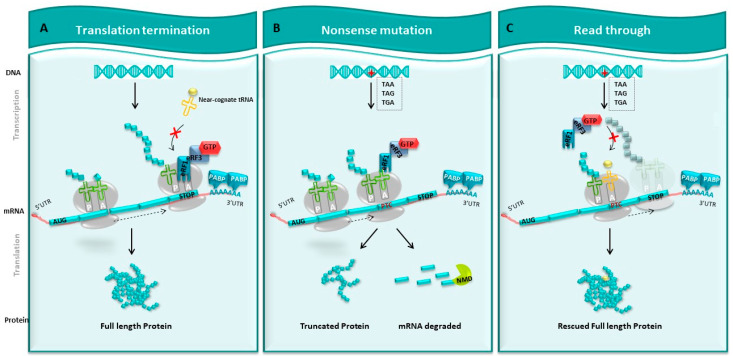
Readthrough strategy. (**A**) Normal translation. DNA is transcribed into mRNA. On this mRNA, there is a normal stop codon at the end, which will be translated into a full-length protein by the ribosome. (**B**) Nonsense mutation. Introducing stop codon (TAA, TAG, TGA), this mutation causes the appearance of a premature stop codon (PTC) that prevents the ribosome from continuing to translate mRNA. This leads to the synthesis of a truncated protein and potentially to the degradation of this mRNA by the NMD pathway. (**C**) Strategy of readthrough. The use of pharmacological molecules promotes the entry of tRNA at the level of PTC during translation, which allows the ribosome to pass through the PTC and continue the translation and re-express a whole protein, sometimes modified from a gene interrupted by a stop codon.

**Figure 3 pharmaceuticals-17-00314-f003:**
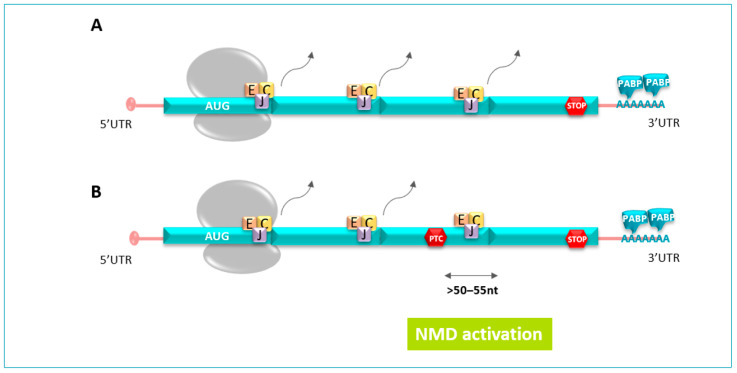
(**A**). During splicing, a multiprotein complex called tje Exon Junction Complex (EJC) is laid 20 to 24 nucleotides upstream. During the translation, the ribosome removes all the EJCs present on the open phase reading until the stop codons. (**B**). PTC, located typically >50–55 nucleotides upstream of the last exon–exon junction, holds back EJCs downstream of the premature stop codon, which are recognized as a signal for the presence of a PTC; therefore, mRNA is targeted for degradation by the NMD pathway.

**Figure 4 pharmaceuticals-17-00314-f004:**
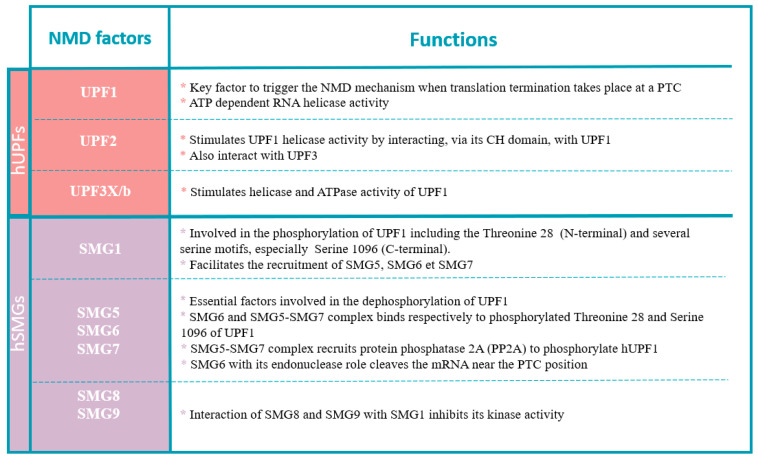
Functions of NMD factors.

**Figure 5 pharmaceuticals-17-00314-f005:**
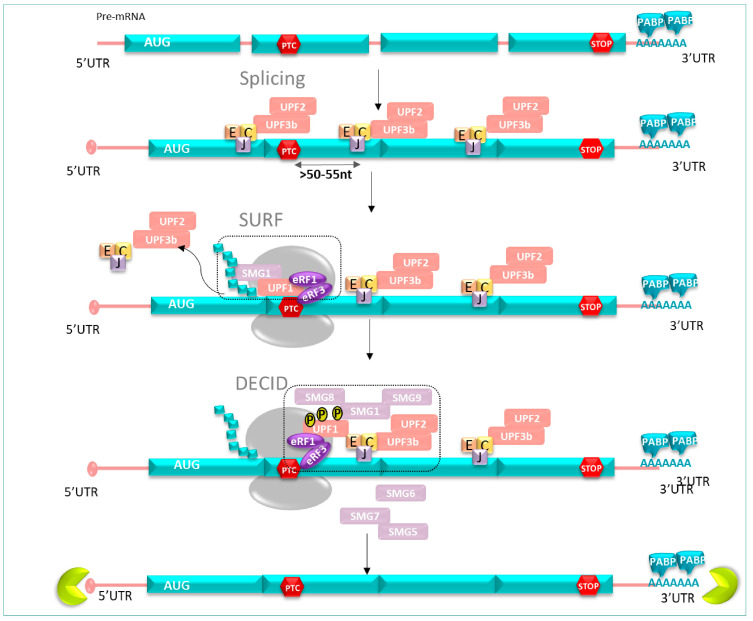
When the ribosome stops at PTC, protein UPF1-SMG1 and eRF1 eRF3 factors form a SURF complex. Then, the SURF complex interacts with the complex EJC-UPF2-UPF3 to form DECID; this allows UPF1 phosphorylation by SMG1 kinase, inducing the recruitment of SMG6 SMG7-SMG5 to dephosphorylate UPF1, and the phosphorylation/dephosphorylation steps set off degradation events.

**Figure 6 pharmaceuticals-17-00314-f006:**
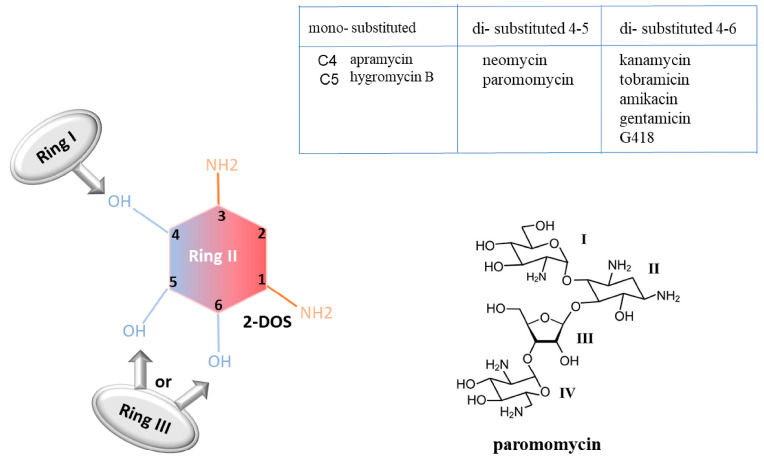
Aminoglycoside chemical structure. Example of aminoglycoside “paromomycin” [55].

**Figure 7 pharmaceuticals-17-00314-f007:**
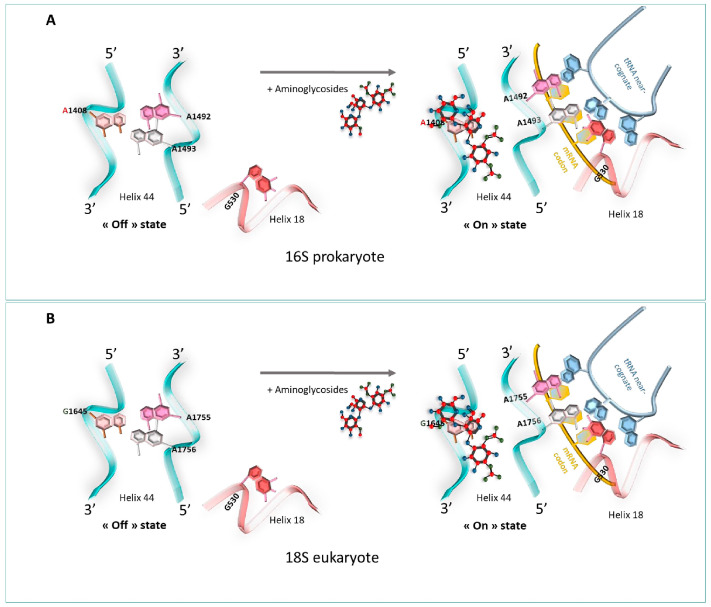
Aminoglycosides’ mechanism of action. (**A**) 16S prokaryote, from the left side of the panel, at the A-site: site of the ribosome. Bases A1492 and A1493 are positioned towards the inside of helix 44 (off state). On the right side, the codon is recognized by the tRNA, and A1492 and A1493 switch outside of helix 44 (on state). Strong fixation of aminoglycosides to A1408 in the rRNA of the A site changes the conformation in which the adenines A1492 and A1493 are directed towards the outside of the helix in the presence of a near-cognate tRNA, inducing a loss of its ability to discriminate between cognate and non-cognate tRNA-mRNA and an inhibition of protein synthesis. (**B**) 18S eukaryote. Low fixation of aminoglycosides to G1645 in the rRNA of the A site displaces the adenines A1755 and A1756 of helix 44, outside of the helix in the presence of a near-cognate tRNA, maintaining the translation process. A1408: high affinity = bacterial death; G1645: low affinity = continuity of translation.

**Figure 8 pharmaceuticals-17-00314-f008:**
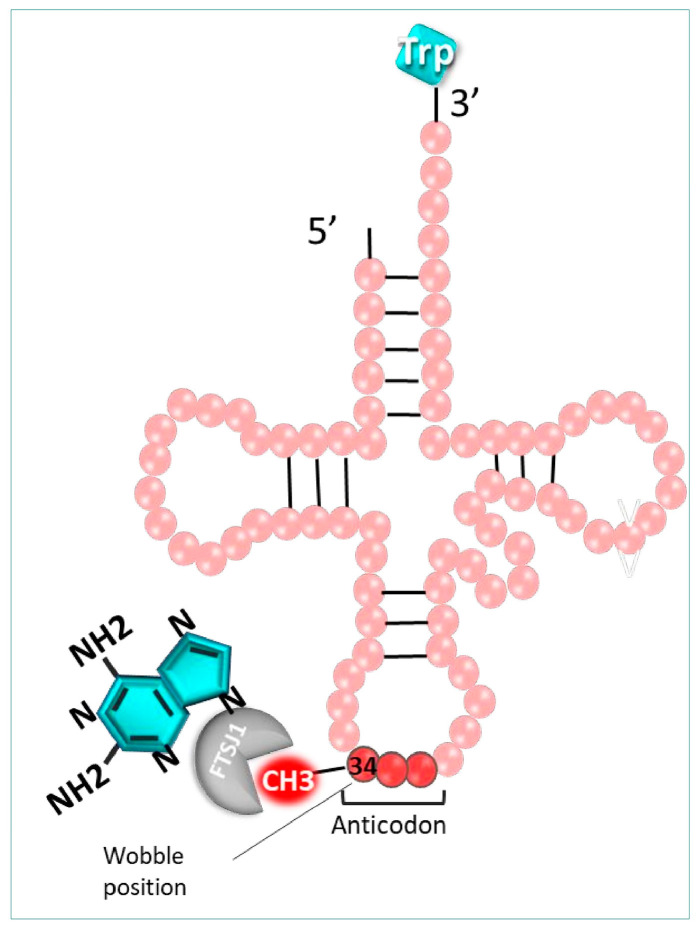
Mechanism of action of 2-Diamin purine. DAD (blue) inhibits FTSJ1 enzyme, which is responsible for the 2′-O-methylation of cytosine 34 of the anticodon in tRNATrp.

**Figure 9 pharmaceuticals-17-00314-f009:**
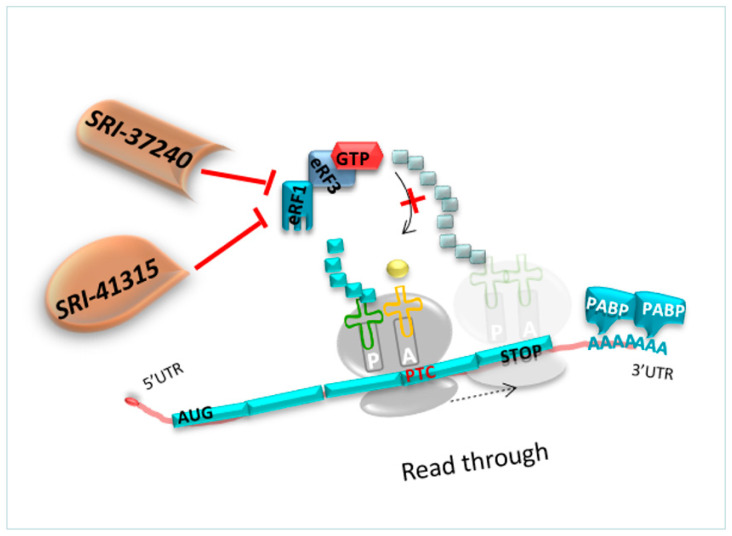
Mechanism of action of molecules targeting eRF1. The molecule SRI-41315 and its derivative SRI-37240. The latter is a promising candidate to block eRF1, paving the way for the incorporation of a near-cognate amino acid at the PTC position.

**Figure 10 pharmaceuticals-17-00314-f010:**
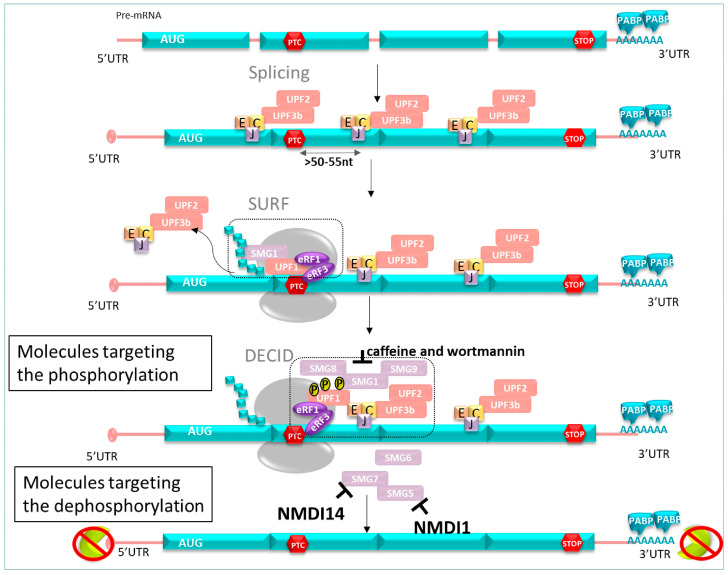
NMD inhibitor pathway. Molecules targeting phosphorylation: caffeine and wortmannin inhibit SMG. Molecules targeting dephosphorylation: NMDI 1 inhibits SMG5 interactions, and NMDI 14 inhibits SMG7 interactions, resulting in the prevention of mRNA degradation harboring PTC.

## Data Availability

Not applicable.

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
