# Peer review of "Readthrough Activators and Nonsense-Mediated mRNA Decay Inhibitor Molecules: Real Potential in Many Genetic Diseases Harboring Premature Termination Codons"

_pharmaceuticals, 2024, doi:10.3390/ph17030314_

Round 1
Reviewer 1 Report
Comments and Suggestions for Authors
This article reiviews nonsense mutations, the therapeutic strategy of readthrough, and the potential applications of readthrough inducers, particularly aminoglycosides. Here's an analysis of the content:
Overview of Nonsense Mutations and Readthrough:
The article introduces the concept of nonsense mutations leading to premature termination codons (PTC) and explains the consequences – either rapid degradation of mutated mRNA or the production of a truncated protein.
It mentions the therapeutic strategy of readthrough, which involves using small molecules drugs to bypass PTC. These drugs facilitate the incorporation of a near cognate tRNA at the PTC position through the native polypeptide chain.
Organized Review of Strategies:
The article indicates that the subsequent review will detail various existing strategies organized according to pharmacological molecule types and their mechanisms. This suggests a structured examination of the strategies employed in addressing nonsense mutations.
Positive Results and Potential Applications:
Positive results observed in testing readthrough molecules in multiple neuromuscular disorders models are highlighted. The implication is that the readthrough approach shows promise for addressing peripheral neuropathies, suggesting potential applications in various diseases.
Introduction of Aminoglycosides:
The article introduces aminoglycosides as the first molecules identified as readthrough inducers. It hints at subsequent studies identifying new molecules that can correct nonsense mutations, potentially restoring the protein of interest.
Broad Application and Importance:
The article emphasizes the potential significance of these molecules in treating a wide range of diseases associated with nonsense mutations, particularly in cases where therapeutic approaches are lacking. It mentions neurological diseases, cancer, and rare genetic diseases.
Call for Personalized Medicine:
The conclusions emphasize the need for a better understanding of the molecular mechanisms behind these strategies. It suggests that this knowledge is crucial for designing molecules tailored for personalized medicine, acknowledging the variability in efficiency based on the specific PTC and genetic background.
Suggestions for Improvement:
Clarify the term "rapide degradation" to improve readability.
Consider using "review" instead of "revue" for the correct term.
Ensure consistency in the use of terminology, such as "readthrough" and "readthrough molecules."
Provide more context or explanations for terms like "NMDI" for readers who may not be familiar with the subject matter.
Overall, the article is informative and provides a good foundation for understanding the significance of readthrough strategies in addressing nonsense mutations.
Author Response
We would like to thank the reviewers and the editor for the time spending on reviewing our manuscript and for the positive comments. We greatly appreciated your valuable feedback and suggestions.
Response to Reviewer 1
Comments and Suggestions for Authors
Response: We thank the reviewer for the time, and the helpful suggestions that improved our review. We corrected the minor errors as follows, highlighted in yellow in the text.
Clarify the term "rapide degradation" to improve readability.
Response: We thank you for this suggestion, and we clarified this point by replacing “Nonsense mutations generating a premature termination codon (PTC) can induce both rapide degradation of the mutated mRNA, or the production of a truncated protein.” by “Nonsense mutations generating a premature termination codon (PTC) can induce both accelerated degradation of the mutated mRNA compared to the wild-type version of the mRNA, or the production of a truncated protein.”
Consider using "review" instead of "revue" for the correct term.
Response: We thank you for caching the error. We corrected it by writing “review” instead of “revue” in the abstract.
Ensure consistency in the use of terminology, such as "readthrough" and "readthrough molecules."
Response: We thank you for noticing this point. The terminology changes depending on the context. We used “readthrough” when it is referred to the process or the mechanism efficacy as “PTC readthrough, readthrough activity”, and we used “readthrough molecules”, when it is related to the compounds that suppress the nonsense mutation.
Provide more context or explanations for terms like "NMDI" for readers who may not be familiar with the subject matter.
Response: We thank you for your comment. We added line 95 the term “(NMDI)” after the term “NMD inhibitors”.

Reviewer 2 Report
Comments and Suggestions for Authors
This review is fully summarized the drugs for PTC treatment in mechanisms and related applications. All writing is good and clear to understand. Some of points were needed to revise as following:
In line 19: “In this revue” should be “In this review”
In line 85: “NMD” should be mentioned as the full name “nonsense-mediated mRNA decay (NMD)” for the first time appearing in the main article.
In line 236: “eIF4A3, MAGOH et Y14, MLN51” should be “eIF4A3, MAGOH, Y14, MLN51”
In line 667-670 should be added the reference about the results of PTC414.
In line 677: “RTC13 et RTC14” should be “RTC13 and RTC14”
In line 695-698 should be added the mechanism of action of RTC14.
In line 712 should be added the mechanism of action of GJ071.
In line 715, “RT13” should be “RTC13”.
In line 801, “SRI-37240 SRI-41315” should be “SRI-37240 and SRI-41315”.
In line 982: “RDEB” should be mentioned as the full name “recessive dystrophic epidermolysis bullosa (RDEB)” for the first time appearing in the main article.
In line 988-1002 should be added a reference.
Author Response
We would like to thank the reviewers and the editor for the time spending on reviewing our manuscript and for the positive comments. We greatly appreciated your valuable feedback and suggestions.
Response to Reviewer 2
Comments and Suggestions for Authors
Response: We thank the reviewer for the time and the helpful suggestions that improved our review. We corrected the minor errors, highlighted in yellow in the text.
In line 19: “In this revue” should be “In this review”
Response: We thank you for caching the error. We corrected it by writing “review” instead of “revue” in the abstract.
In line 85: “NMD” should be mentioned as the full name “nonsense-mediated mRNA decay (NMD)” for the first time appearing in the main article.
Response: We thank you for noticing this point. The full name “nonsense-mediated mRNA decay (NMD)” has been added for its first appearance in the main article in line 87.
In line 236: “eIF4A3, MAGOH et Y14, MLN51” should be “eIF4A3, MAGOH, Y14, MLN51”
Response: We are sorry for this mistake. We fixed it.
In line 667-670 should be added the reference about the results of PTC414.
Response: We thank you for your comment. In fact, the different results described in this paragraph presenting the results of PTC414 on fibroblasts but also on zebrafish model were from the same publication written by Moosajee et al in 2016. We added the reference number of this article in line 673.
In line 677: “RTC13 et RTC14” should be “RTC13 and RTC14”
Response: We thank you for caching this error. The change was performed, written “RTC13 and RTC14”.
In line 695-698 should be added the mechanism of action of RTC14. In line 712 should be added the mechanism of action of GJ071.
Response: We thank you for noticing that we did not precise it in the first version of the manuscript, but the molecular mechanisms of action of RTC14 and GJ071 remain unknown.
We modified the text, line 699 to add this information on RTC14 by writing “The molecular mechanism that lies behind RTC13 and RTC14 readthrough activity effectiveness remain unknown. These molecules possibly display a similar mechanism of action as aminoglycoside, interfering with ribosomal translation.”.
We also added this information concerning GJ071 by writing, in line 715“The molecular mechanism of GJ071 remains unknown.”
In line 715, “RT13” should be “RTC13”.
Response: We thank you for caching the error. The correction has been performed.
In line 801, “SRI-37240 SRI-41315” should be “SRI-37240 and SRI-41315”.
Response: This word was indeed missing. We added “and” between the two molecules’ name.
In line 982: “RDEB” should be mentioned as the full name “recessive dystrophic epidermolysis bullosa (RDEB)” for the first time appearing in the main article.
Response: We thank you for noticing this point. The full name “recessive dystrophic epidermolysis bullosa” has been added.
In line 988-1002 should be added a reference.
Response: We thank you for noticing this point. We added the references of Mendell et al, 2004 and of Wittmann et al, 2006 (references number: 153, 154).